# Attitudes, Risk Factors, and Behaviours of Gambling among Adolescents and Young People: A Literature Review and Gap Analysis

**DOI:** 10.3390/ijerph18030984

**Published:** 2021-01-22

**Authors:** Ben J. Riley, Candice Oster, Mubarak Rahamathulla, Sharon Lawn

**Affiliations:** 1College of Medicine and Public Health, Flinders University, Adelaide, SA 5042, Australia; ben.riley@sa.gov.au (B.J.R.); candice.oster@flinders.edu.au (C.O.); 2Faculty of Health and Medical Sciences, Social Work, University of Western Australia, Perth, WA 6000, Australia; mubarak.rahamathulla@uwa.edu.au

**Keywords:** gambling, adolescent, youth, problem gambling, review

## Abstract

Gambling is occurring in a rapidly changing landscape, with new trends and technologies affecting gambling behaviour and problem gambling across a range of populations. Gambling activity among adolescents and young people has received considerable research attention due to a high prevalence of gambling reported among these groups in recent years. Despite legislation worldwide to constrain gambling among adolescents and young people, modern technology, such as online gaming apps and online gambling venues, has significantly increased their exposure to the risks of problem gambling. It is important, therefore, to have up to date information about what is currently known about gambling and to explore gaps in our knowledge. This gap analysis presents the results of a systematic approach to reviewing the current literature on gambling behaviour, attitudes, and associated risk factors for gambling and problem gambling among adolescents and young adults (aged 10–25 years). The review included studies published between January 2015 and August 2020 and included 85 studies for final synthesis. Findings reveal further research is needed on the implications for young people of emerging technologies and new trends in gambling in the digital age. The current gap analysis reveals that this should include more research on the development and impact of both treatment and intervention strategies, and policy and regulatory frameworks from a public health perspective.

## 1. Introduction

Gambling activity among adolescents and young people (under 25 years) has received considerable research attention due to a high prevalence of gambling reported among these groups in recent years. A recent systematic review reported that 0.2 to 12.3% of youth met criteria for problem gambling [1], with some researchers predicting the prevalence of problem gambling among adolescents may be comparable to that of adult populations [2]. Despite legislation worldwide to constrain gambling among adolescents and young people, modern technology, such as online gaming apps and online gambling venues, have significantly increased their exposure to the risks of problem gambling [3]. Moreover, there is emerging evidence that the COVID-19 pandemic has led to an increase in engagement with online gambling [4,5,6], and while the populations surveyed to examine gambling during the pandemic to date have involved individuals aged 18 or over, those who indicated gambling problems were more likely to be younger [4,6].

Gambling appears to be exceedingly common among adolescents. For example, Rasanen et al. [7] reported that as many as 50–80% of Nordic adolescents gambled in the past year, despite it being illegal to gamble in most Nordic countries before the age of 18. Similar trends have been reported in other parts of the world, such as Canada, the United States, the United Kingdom, and Australia, where it is reported that an average of 60–80% of young people aged 13–17 years gambled at least once per year, with 3–5% of adolescents displaying signs of problem gambling [8]. Gambling is now one of the most frequently reported addictions among young people [9].

Adolescent problem gambling can lead to many complex problems, such as criminal behaviour, poor academic achievement, school truancy, financial problems, depressive symptoms, suicide, low self-esteem, deterioration of social relationships, and substance abuse [10]. More than two thirds of adult gamblers have reported that exposure to gambling during adolescence was a key contributing factor to their current gambling [11], indicating that preventive measures taken to reduce gambling during adolescent years could potentially reduce the prevalence of problem gambling in the adult population. Understanding young peoples’ attitudes towards gambling, how their attitudes are influenced, and the reasons why they gamble, despite it being illegal for underage youth in most jurisdictions, will assist researchers and policy makers in considering appropriate actions.

Gambling is occurring in a rapidly changing landscape, with new trends and technologies affecting gambling behaviour and problem gambling across a range of populations [12]. It is important, therefore, to have up to date information about what is currently known about gambling and to explore gaps in our knowledge [13]. The aim of this review, therefore, is to summarise the evidence published during the previous five years pertaining to adolescents’ and young adults’ attitudes to gambling, along with associated risk factors, and identify gaps in our understanding, particularly regarding emerging technologies. Specifically, our research question was: what are the gaps in our understanding of attitudes and behaviours towards gambling among adolescents and young adults (under 25 years), and the associated risk factors?

## 2. Materials and Methods

The most commonly accepted approach to reviewing the literature is by conducting a systematic review, with the purpose of identifying and synthesising scholarly research on a topic. While it is important to understand the evidence relating to a particular concept or phenomenon, identifying gaps in our understanding is necessary in order to develop a research agenda to advance knowledge [14]. This review, therefore, focuses on both summarising the state of the evidence and synthesising gaps across the body of the literature to inform future research.

There are a number of approaches to conducting a gap analysis [13]. We adopted a systematic approach, modified from Otto et al. [15], as follows:Stage 1: Systematic literature search of peer-reviewed literature published since 2015Stage 2: Summary of areas of research focus identified in the literatureStage 3: Identification of gaps as stated within each articleStage 4: Identification of gaps across the body of researchStage 5: Identification of further gaps not identified in the body of research.

The search strategy was developed in consultation with a research librarian. A systematic search was conducted of the electronic databases Medline, Emcare, PsycINFO, SCOPUS, Web of Science, and Proquest (Health & Medicine, Social Sciences Collection). The search terms included “Gambling”, ”Betting”, ”Wagering”, ”Pokie”, ”Lottery”, ”Casino”, ”Keno”, ”Machine”, ”Video Games”, ”Technology or Information Technology”, ”Trend/emerging trend/future/interactive/innovation in Gambling Technology”, and ”Computer games”.

We chose to limit the review to papers published since 2015 to ensure the currency of the review and gap analysis, mindful that any included literature reviews would capture earlier studies within their synthesis. This concern for currency was exemplified and verified in our initial reading of earlier reviews, such as that by Blinn-Pike et al. [16], which did not emphasise or even mention the issue of internet/phone usage and gambling by young people at the time of its publication. Clearly, the growing concern about internet gambling in this population has only become apparent in more recent years. The search was done in two phases. The first search was conducted on 16 October 2018 as part of a rapid review funded by the Office of Responsible Gambling (New South Wales, Australia). We then conducted an updated search on 26 August 2020 using the same search strategy. We did not limit the search to youth or young adults, as there was a risk of missing some relevant articles that may not necessarily include those terms.

The search results were uploaded to Covidence screening and data extraction software for screening. After removal of duplicates, three reviewers (the first two authors and the last author) conducted the screening process. The reviewers were two senior researchers with previous experience in conducting several literature reviews, and a senior researcher with specific expertise in gambling treatment and research with some prior experience in conducting literature reviews. The inclusion and exclusion criteria for the current review (Table 1) were then applied to the 116 articles acquired from the original search and the 88 articles acquired from the updated search, plus five papers identified during peer review. This resulted in a combined total of 85 articles identified for the current literature review and gap analysis. The results of the search are presented in the PRISMA diagram in Figure 1.

Quality appraisals were conducted using the Mixed Methods Appraisal Tool (MMAT) for qualitative and quantitative randomised controlled trials, quantitative non-randomised trials, quantitative descriptive studies, and mixed methods studies [17]. The purpose of the quality appraisal was to identify any gaps in the quality of the evidence, rather than exclude low-quality research. However, we determined that all peer-reviewed research was generally well-conducted and of good quality. The appraisals showed that the great majority of articles reviewed were quantitative descriptive studies. In measuring problem gambling, most used existing validated scales designed for adults, not specific to adolescents. Several contained samples where cultural context likely impacted outcomes due to different parenting or family expectations. Furthermore, as most studies were descriptive, the underlying mechanisms and/or temporal relationships between variables were not examined. See Appendix A for details.

## 3. Results

We identified 85 articles on gambling among adolescents and young adults comprised of two experimental, one pre-post, five longitudinal, and 68 cross-sectional studies, along with one commentary and six literature review articles. The 85 articles spanned 23 countries, with USA and Italy containing the most (*n* = 8), followed by Spain, Australia, Finland, Canada (*n* = 6), the UK, and Hong Kong (*n* = 4). Greece produced three articles, and the remaining countries of Nigeria, Portugal, Sweden, Korea, Croatia, Poland, Finland, Ethiopia, Israel, China, India, Denmark, Ghana, Norway, and Germany produced one or two articles. The age range of young people across most studies was 10 to 25 years, with one qualitative study involving a sample aged 15 to 35 years [18], and reported the age of participants alongside the data. The majority of studies (*n* = 46) involved secondary school students, with the remainder involving university or college students, or young people within a community sample; one study involved an adult problem gambling treatment-seeking population and examined their gambling during adolescence [19]. See Appendix A, Appendix A for a summary of the included articles.

### 3.1. Gambling Participation and Problem Gambling

Most studies observed whether young people had ever participated in gambling or had gambled within the previous 12 months, and reported the prevalence of problem gambling, which was typically reported as low or moderate risk of problem gambling, or problem gambling. Lifetime participation in gambling rates ranged between 42.1% [10] and 89.9% [20], with the majority of studies that examined gambling participation reporting that around a third or more of adolescents or young adults confirmed that they had gambled at least once in their lifetime [10,20,21,22,23,24,25,26,27,28,29,30,31,32,33,34,35,36,37]. Gambling participation rates during the past 12 months ranged between 18.6% [38] to 85% [23]. Differences between gambling participation rates were suggested to be in part attributed to whether all gambling activities, such as playing cards with family, the lotto, or private wagering with peers, were included [23,35], and whether the study population involved lower sociodemographic characteristics, for example unemployment [19] or being from developing countries, whose youth may have a greater attraction to gambling as a means to overcome poverty [39]. Problem gambling prevalence rates ranged between 1.1% [10] to 9.8% [40], with the majority of studies reporting ranges between 3.6 to 5.6% [23,41,42,43,44,45]. Older (16–19 years) adolescents were more likely to have problems with gambling than younger (13–15 years) adolescents [46]. A recent literature review of qualitative studies on adolescent gambling reported on the normalisation of gambling among youth and its embeddedness in everyday life [47].

### 3.2. Risk Factors

Given that almost all of the studies were cross-sectional by design, they were unable to infer anything about temporal relationships between associated risk factors. However, there were several key risk factors associated with problem gambling in adolescents and young people that were consistently reported across studies. The most frequently reported associated risk was being male, followed by the attitudes of parents, family, and friends towards gambling, involvement with alcohol and/or other substances, sensation seeking, and poor social connectedness.

#### 3.2.1. Young Males

Young males were reported to have both greater participation in gambling and gambling problems than females [10,20,21,23,25,30,31,32,33,34,35,36,46,48,49,50,51,52,53,54]. Indeed, being male was reported to increase the odds of being a moderate to high risk problem gambler by 25 [19] to 37 times [44]. In addition to gambling for money, males were also more likely to have participated in simulated gambling [51]. Poor school grades were associated with problem gambling for males and females [25,29,32], however, older male adolescents who struggled academically and whose fathers had low education levels were particularly at risk [33]. Male gamblers were more likely to engage in multiple forms of gambling than female gamblers [23].

Socioemotional harms associated with gambling were positively correlated with age for both male and female adolescent gamblers, and this interaction was significantly more pronounced for males [35]. Furthermore, although a significant relationship between involvement in competitive sports and gambling frequency was reported for both genders, it was associated with problem gambling only for males [51]. However, while males were more likely to be frequent gamblers than females, female frequent gamblers were more likely to have problems with gambling than frequent gambling males [46]. A longitudinal study by Pallesen et al. [54] found that males were particularly influenced by parental attitudes towards gambling: parental approval of gambling behaviour at age 17.5 years predicted acceptant attitudes towards gambling for male and female adolescents at age 18.5 years, and this effect was greater for males.

#### 3.2.2. Family and Friends’ Attitudes towards Gambling

A key factor was whether family members or friends gambled or displayed positive attitudes towards gambling. Adolescents were more likely to gamble if they had a friend or relative who either had a gambling problem [22,55,56] or engaged in gambling [10,22,49,57]. Researchers suggested that parents may encourage boys more than girls to be involved with gambling due to cultural influences, which may be one reason why male adolescents are at particular risk of developing gambling problems [41]. Adolescents whose parents gambled had significantly more positive attitudes towards gambling than those with non-gambling parents [22], and those with family members or friends who gambled were more likely to report that they were either currently gambling [22], had gambled during the past year [23], or had a gambling problem themselves [50]. Having a parent who gambled was related to delinquent behaviour, along with adolescent gambling involvement [58], and having a personal relationship with anyone who gambled increased the likelihood of the adolescent having a gambling problem [10]. In addition to being influenced by their attitudes, adolescents reported being assisted with gambling by family and friends. In one study, over a third of adolescents reported that they had placed wagers via family and friends [56], while another observed that adolescent gambling was usually facilitated by a parent, particularly with access to scratch-it tickets and sports betting [53]. On the whole, social factors, such as gambling involvement by family and friends, have been shown to play a greater role in adolescent gambling participation than psychological factors [58].

#### 3.2.3. Alcohol and/or Other Substances

Among problem gambling adolescents and young people, there was a high prevalence of engagement with alcohol and/or other substances. Problem and at-risk gamblers, compared to non-problem gamblers, were more likely to consume alcohol [30,59], tobacco [30,60], experience alcohol-related problems [30,61,62,63,64], and use sedatives [63]. Both male and female adolescents who reported gambling during the previous 12 months had greater drug use and involvement with violence than non-gamblers [38]. Interestingly, lower-risk problem gamblers were more at risk of harmful alcohol and/or substance use than high-risk problem gamblers [20,43], with researchers suggesting this may be because high-risk gamblers have less money available to purchase substances [43].

#### 3.2.4. Sensation Seeking

There was consensus among the literature that sensation or excitement seeking was a strong predictor of problem gambling among adolescents and young people [24,36,40,43,64,65,66,67]. Problem gamblers were reported to be more likely to focus on immediate outcomes than non-problem gamblers [40], and male problem gamblers compared to non-problem gamblers were found to be more focused on the present than on past or future events [68]. Adolescent gamblers who reported gambling to seek excitement were more likely to be at risk for problem gambling, consume alcohol, and possess more gambling permission-giving cognitions [64], and sensation seeking was found to be associated with comorbid problem gambling and heavy episodic drinking among male and female adolescents [65]. While sensation seeking was shown to be associated with problem gambling, adolescents with high sensation seeking traits were more likely to have gambling problems if they were immigrants, suggesting that adolescent problem gamblers should not be considered a homogeneous group [24].

#### 3.2.5. Social Connectedness

Social connectedness both inside and outside the family was associated with problem gambling in adolescents and young people. Poor parental attachment was associated with problem gambling and internet and video game addiction among adolescents [69,70], and this relationship was mediated by alexithymia (difficulty identifying and expressing emotions) in adolescents and young people [70]. Another study found that adolescents and young people with poor attachment to both parents and their peers were more likely to have gambling problems [63], and poor school connectedness was associated with gambling participation in adolescents [35]. Lower family connectedness was associated with adolescent problem gambling, and adolescents with gambling problems reported greater family concerns than their non-problem gambling peers [71]. Being an only child was also identified as a risk factor for problem gambling in adolescents and young people [49].

#### 3.2.6. Other Risks for Gambling

A range of other associated risk factors for adolescents and young adults to commence gambling, and potentially progress to problem gambling, were reported among the studies. Older age, lower parental education, absence of siblings, lower grades, and lower age when first gambled were all associated with risky gambling [41]. A higher family income was associated with adolescent and youth gambling [59], and childhood exposure to tobacco smoke was reported to predict an 18% increase in problem gambling by age 12 [72]. Increased accessibility to gambling venues (e.g., located close to homes) was related to increased problematic gambling among young people [10], as was the participation in more than one form of gambling [55] and the presence of cyberbullying [36]. One study examined the risks associated with different types of gambling among young people, and found that online gambling that involves perceived skill (e.g., online poker, online casinos, bingo) was associated with greater risk than non-skill-based forms (e.g., online slots, lotteries) [73].

Engagement in any sport-relevant gambling activity was a predictor of problem gambling risk, with one study of almost 7000 adolescents [46] and another comprising over 10,000 adolescents [25] reporting sports betting as the most common gambling activity. Experimental research has found implicit associations between gambling and sports, particularly sports that are generally associated with gambling, among male and female adolescents, although the implicit associations were not related to the intention to gamble [74]. Among adolescent gamblers, the level of harm increased with age, and this interaction was more pronounced for males [35]. Hyperactivity, conduct problems [44], emotional and attentional problems, delinquent behaviour [75], and social dysfunction [45] were also related to adolescent problem gambling.

While involvement with online gambling appeared to be less common among adolescents than land-based gambling [21,55,68], those who did gamble online were reported to be at greater risk of problem gambling [25,50]. However, one study of more than 2000 adolescent students reported that internet gambling was not predictive of problem gambling [76]. Participation in social or simulated online gambling was reported to be a key gateway to adolescent and youth online gambling [77,78], and tended to be unsupervised by parents, particularly among males [53], which is notable given that a recent study of 16-year-olds found that parental monitoring was a protective factor against problem gambling [79]. However, qualitative research observed that, although online social gambling led some adolescents to gambling online with money [80], for others, it reduced the likelihood that they would wager with real money [78].

### 3.3. Attitudes towards and Reasons for Gambling

One study focused directly on adolescents’ attitudes towards gambling, and found that the majority of respondents viewed gambling as a risky activity, with less than a third viewing it positively, for example as a quick way to make an income [39]. However, more than half of the young respondents believed that gambling yields a high financial return on their investment [39]. In an experimental study, adolescent problem gamblers displayed greater bias towards attending to gambling-related cues that promoted the potential for financial gains than their non-problem gambling peers [81], and another study found that adolescents’ attitudes towards money and distorted gambling cognitions mediated the relationship between social cynicism and fate control on gambling participation [82].

From the studies that examined the reasons why adolescents and young people participate in gambling, two key reasons emerged: to make money and to regulate emotions. In countries containing populations with low socioeconomic populations, young people had a vulnerability to being attracted to ways to overcome poverty, and gambling was conceived by adolescents as one such way [22,40]. For instance, unemployed youth described gambling as an easy means to make money for their daily needs [18]. At-risk and problem gambling adolescents were more likely to gamble to win money than not at risk problem gamblers [23], and more males gambled for excitement than females [64].

Gambling as a means for adolescents and young people to manage unwanted emotions was a frequently reported motivation for gambling. Problem gamblers scored higher on emotional dysregulation and maladaptive coping strategies [70], and difficulties in emotional regulation mediated the relationship between gambling motives and gambling severity [62]. Higher scores on problem gambling were associated with motivation to increase positive emotion [62], and those acknowledging self-harming behaviours were more likely to report at risk or problem gambling and more permissive gambling cognitions [83]. Moreover, young adult gamblers viewed gambling as a positive social activity [84], and reported that it helped them to manage stress [18].

### 3.4. Protective Factors

Social connectedness appeared to be the strongest protective factor. A positive relationship with parents and engagement in meaningful leisure time activities, such as organised extracurricular activities, were reported to be protective factors against gambling involvement [44]; so too was involvement in prosocial behaviour, which was negatively associated with problem gambling [44]. Adolescents who perceived their parents to have greater knowledge about their whereabouts and greater knowledge about gambling-related harms were less likely to gamble [85]. Social support from parents and schools was related to a reduction in gambling among boys and girls [86].

Online socialisation was also examined with respect to problem gambling. Adolescents and young people who identified more strongly with offline peer groups were less likely to have gambling problems than those who socialised online [87]. Those who engaged with online simulated gambling reported that social interaction was a key motivator [78].

One study specifically examined reasons for not gambling among young people [88], and found that several key reasons mirrored common reasons why young people do gamble. For example, concerns that friends or family would judge them negatively, knowledge about the odds, and competing activities or priorities were frequently reported as reasons not to gamble. Parental discipline and adolescent coping styles protected at-risk infrequent gamblers, but not adolescents with high gambling involvement [89].

### 3.5. Vulnerable Populations

Only three studies specifically examined gambling among vulnerable populations. Immigrant adolescents reported higher levels of problem and at-risk gambling than non-immigrants [24,27]. Another study examined gambling participation among out-of-school adolescents (students who had left school for various reasons, such as to enter vocational training, were living in shelters, or suffered from diseases). The authors argued that most adolescent gambling studies involve samples recruited from secondary schools, yet many vulnerable youths do not go to school. The study found that almost two thirds of out-of-school adolescents reported gambling within the past year, and 40% during the past three months, with the most common motivation for gambling being to gain excitement (42.9%), followed by to win money (34.1%) [31]. Another study assessed gambling behaviour among transgender diverse and cisgender populations. Both male and female assigned at birth transgender adolescents had higher problem gambling than cisgender males and females [37].

### 3.6. Identification of Gaps Noted within Existing Research

The most frequently reported gap across studies was a need for the development of treatment and preventative programs, along with efficacy studies [21,29,38,39,62,63,64,90], including specific programs that target subgroups of adolescents and young people whose needs may be different, such as low-, moderate-, and high-risk gamblers [31], at-risk adolescents and young people [43], and different age groups [66]. Studies were in agreeance that preventative and educational programs should begin in secondary schools [26,27,88], particularly because 14 years was the most frequently reported age of gambling initiation among adolescents [33]. The development and investigation of intervention programs involving families and the wider community was also recommended [84].

The next most frequently reported gap raised by authors of the reviewed articles concerned public health and regulatory issues, particularly around the online environment and new and evolving forms of gambling [76,91], for instance, fantasy sports [25], and the impact of gambling advertisements on young people [74,92]. Although the research is still young, evidence supporting a link between online social gambling and adolescent gambling participation and problem gambling is growing. Several researchers called for further research into the way adolescents interact with online social gambling operators, including within the context of video games [25,34,53,77,78,80].

A need for research on the differences between distinct forms of gambling was reported [39,73], including different modes of access [55], along with more research on the role of parental attachment [69]. Research on the impact of non-gambling games, such as loot boxes, is in its infancy, and there appears to be some disagreement concerning the similarities and differences between gambling and non-gambling games [56,93]. Clearly, further research is needed in this area, particularly given the growing prevalence of loot boxes, with 58% of the top mobile and desktop games on the Google Play store containing them [94].

### 3.7. Identified Gaps across the Body of Research

Longitudinal studies are needed to establish causal factors of the key risk factors identified in the research to date. Most of the research appears to be focused on the individual level, and, to a large extent, ignores broader socio-political factors, although the accessibility/availability of gambling products was raised by one study [10]. Research that examines, for example, the influence of legislation and gambling marketing and advertising is needed. Additionally, further research is needed on vulnerable populations of adolescents and young people, such as cultural minorities and adolescents not attending school.

Both quantitative and qualitative research is needed. Population surveys can measure the extent of a problem and monitor trends over time, while qualitative methods allow a deeper exploration of issues. For example, population-based surveys have indicated that engagement with online social gambling predicts adolescent gambling and risky gambling (e.g., King et al. [53]). However, a qualitative study by Kristiansen et al. [78] revealed that, for some adolescents, participating in online social gambling made them less likely to gamble with real money. Understanding how young people engage with and are influenced by social gambling will help inform the development of harm prevention strategies.

## 4. Discussion

The purpose of this literature review was to summarise the current knowledge and identify gaps in our understanding about gambling behaviour and attitudes among adolescents and young adults. Overall, this review demonstrates the breadth of focus in the gambling literature identified since 2015, reflected in the proliferation of cross-sectional research. This suggests that abundant descriptive work has been performed across the various domains, with very little empirical focus on treatment and intervention strategies, both from the individual and public health levels. What is needed now is a research agenda focused on depth, building sequentially on previous research to best support the development of prevention, harm minimisation, and treatment efforts.

Research has tended to focus on gambling behaviour (particularly the prevalence of gambling and problem gambling), with very limited recent research exploring attitudes towards and reasons for gambling among adolescents and young adults. However, there is an early indication that gambling by adolescents and young people may be driven by motivations to relieve stress [18] and interact socially [84], which is further validated by research that reports that engagement in extracurricular activity is a key protective factor [44]. This is particularly important concerning adolescent online gambling. The research to date suggests that adolescents may be attracted to online simulated gambling sites for social interaction reasons [77,87], and simulated gambling is a gateway to wagering for real money [77,78] and tends to be unsupervised by parents [53]. Further research is needed to understand the motivation to gamble among adolescents and young people, along with their parents’ attitudes and understanding of the accessibility of evolving forms of gambling. International research reports regular gambling participation and prevalence of problem gambling among adolescents and young adults. However, there is limited recent research into the prevalence of gambling behaviour among vulnerable adolescent populations. This has important implications for the prevention and early intervention efforts with these populations.

Longitudinal research is needed to explore potential changing attitudes, particularly regarding Internet gambling, simulated gambling, and motivators for gambling. Smartphone sports bettors have reported that the immediate accessibility and the pervasiveness of online sports betting marketing make it challenging to control sports betting involvement [95], and research has shown that marketing plays an important role in the normalisation of gambling in sports [96]. Furthermore, sports bettors have indicated that the saturation of marketing for sports betting has normalised gambling in sports [96]. The evidence to date clearly suggests that sports betting is one of the most common forms of gambling among adolescents [25,46]. Accordingly, there is a need for more research on the influence of gambling marketing and timing of advertising, particularly related to sports betting, on children, adolescents, and young people.

### Limitations

This research has several limitations that should be considered while interpreting the findings. Given the focus on contemporary literature for the purposes of gap analysis, a key limitation relates to the omission of literature published prior to 2015. Furthermore, we did not search reference lists. Only peer-reviewed articles written in English were included. Information from books, conferences, unpublished work, or grey literature was not included. As such, there may have been relevant studies in the grey literature, such as reports from major gambling research bodies. A more general limitation likely arises from the process of identifying gaps from the existing body of literature, relying on the gaps that research teams identified as important. This may account, for example, for the limited research with certain subpopulations known to be at greater risk of gambling harm, such as Indigenous populations and culturally and linguistically diverse or refugee populations [16]. Further potential limitations were the chosen search terms and databases. For example, the chosen search terms did not include terms such as “intervention”, “randomized”, “trial”, “evaluation”, or “program”.

## 5. Conclusions

This gap analysis presents the results of a systematic approach to reviewing the current literature on gambling behaviour, attitudes, and associated risk factors for gambling and problem gambling among adolescents and young adults. It is clear that further research is needed on the implications for young people of emerging technologies and new trends in gambling in the digital age. The current gap analysis reveals that this should include more research on the development and impact of both treatment and intervention strategies, and policy and regulatory frameworks from a public health perspective. Research is needed on gambling involvement among vulnerable subpopulations of adolescents who may not be picked up in school surveys, such as those in protective care or in out-of-school learning environments. Finally, more longitudinal research is required to better understand changing attitudes towards gambling and how they are influenced among emerging adults.

## Figures and Tables

**Figure 1 ijerph-18-00984-f001:**
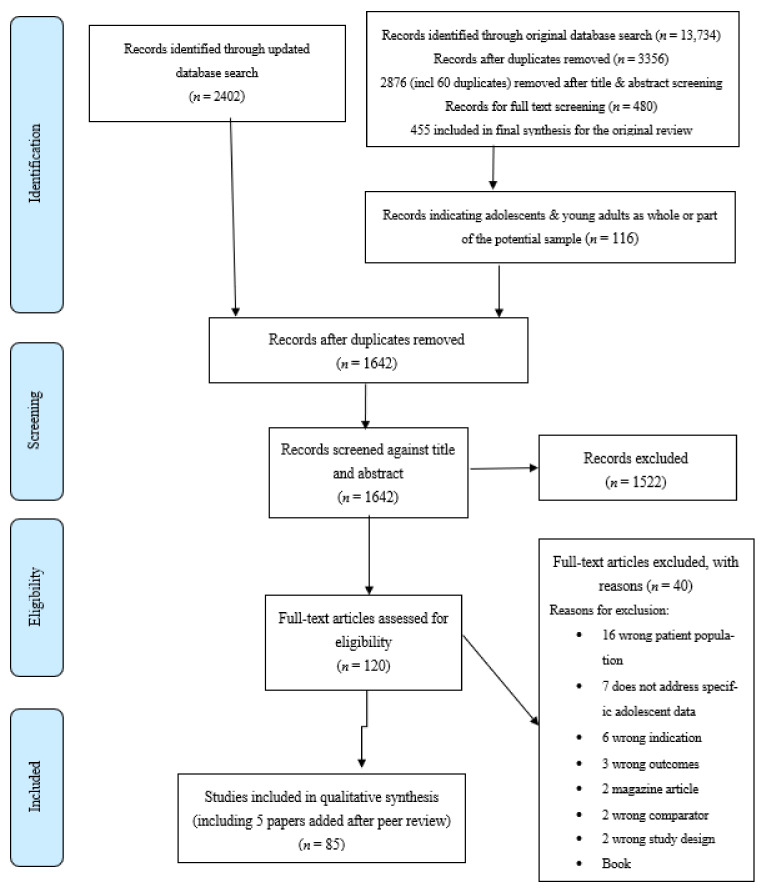
PRISMA diagram.

**Table 1 ijerph-18-00984-t001:** Inclusion and exclusion criteria.

Inclusion Criteria	Exclusion Criteria
English language	Articles not in English
Peer-reviewed articles	Books, conference presentations, PhD theses/dissertations, PowerPoint presentations and posters, government reports
The article reports on an empirical study, systematic review, or review/commentary/discussion article addressing the topic	Study protocols
Published 1 January 2015–26 August 2020	Published prior to 2015
The focus of the article is on gambling, and addresses adolescent and/or youth (10–25 years) * attitudes and/or behaviours towards gambling	The article does not address adolescent and/or youth attitudes and/or behaviours towards gambling, or does not include young people and older adults, but reported only aggregated data
Studies of any quality according to quality appraisal	Only reported on prevalence/incidence

* Articles that included older adults but reported adolescent or young people’s data separately are included.

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
