# Peer review of "Attitudes, Risk Factors, and Behaviours of Gambling among Adolescents and Young People: A Literature Review and Gap Analysis"

_ijerph, 2021, doi:10.3390/ijerph18030984_

Round 1

Reviewer 1 Report

This is timely paper reviewing peer-reviewed literature on attitudes, risk factors and behaviours of gambling among adolescents and young people. The literature search was focused on articles published between 2015 (January) -2020 (August). This study complements well the previous reviews of adolescent gambling (see i.e. Blinn-Pike et al., 2010; Floros 2017). Although I found this paper interesting, there still remain questions and remarks which need authors’ careful attention. The greatest concern for me here is that I am not fully convinced with the literature search process. For some reasons, several relevant peer-review papers published in the field have not been found/ included but met the inclusion criteria as outlined by the authors.

My remarks and comments are listed below.

-Figure 1.  PRISMA flowchart illustrates the number of records found through database searching (original n=116, and updated n=2402). I wonder how it is possible that an updated search found much more records than the original one. This may need some clarification in the text.

-The authors did not tell in the manuscript text how they handled other reviews regarding adolescent gambling published in the same period.

-I strongly suggest that some basic information of the review process is already offered for the readers in the abstract section such as time frame and age range for the review, as well as number of studies included.

-Please state the rationale for choosing the year 2015 for the beginning year of this review.

-In 2010, the Journal of Adolescent Health published a review article (Blinn-Pike et al.) with the purpose of summarizing the research on adolescent gambling. This is worth of mentioning at least in the Discussion section and the authors may want to say something about how their current review complements this.

-It is highly important that the authors emphasize several limitations of their review. For example, potential limitations in the chosen search terms and databases need to be discussed.

-Based on their review, the authors conclude that majority of gambling research has been cross-sectional, while limited focus has been on treatment and intervention research. On the other hand, chosen search terms did not include terms such as “intervention”, “randomized” “trial”, “evaluation” “program” etc.

-Regarding my main critique toward lack of literature search, I have listed examples of several articles which met the inclusion criteria used by the authors but were not included in the review/ were missed. The authors may want to reconsider their review search strategy and search terms in order to include more relevant articles and thus, reduce potential bias in their review. Of course, reviews never could be exhaustive but there are many possibilities to try to improve startegy.

Examples of the articles that met the inclusion criteria but were not found/included in the review:

Molinaro S et al. (2018) Prevalence of youth gambling and potential influence of substance use and other risk factors throughout 33 European countries: first results from the 2015 ESPAD study. Addiction 113, 10.

Spångberg, J & Svensson J. (2020 June) Gambling among 16-year-olds and associated covariates: A Nordic comparison. Scandinavian Journal of Public Health.

Fröberg, F., Rosendahl, I. K., Abbott, M., Romild, U., Tengström, A., & Hallqvist, J. (2015). The incidence of problem gambling in a representative cohort of swedish female and male 16–24 year-olds by socio-demographic characteristics, in comparison with 25–44 year-olds. Journal of Gambling Studies, 31(3).

Raisamo, S., Kinnunen, J. M., Pere, L., Lindfors, P., & Rimpelä, A. (2020). Adolescent Gambling, Gambling Expenditure and Gambling–Related Harms in Finland, 2011–2017. Journal of Gambling Studies, 36(2), 597–610.

Kryszajtys D et al. (2018). Problem gambling and delinquent behaviours among adolescents: A scoping review. Journal of Gambling Studies.

Räsänen T, Lintonen T, Tolvanen A, Konu A. The role of social support in the association between gambling, poor health and health risk-taking. Scand J Public Health 2016: 44 (6).

Oksanen et al. (2019). Gambling patterns and associated risk and protective factors among Finnish young people. Nordisk Alkohol Nark 36(2).

Reviewer 2 Report

Thank you for the opportunity to review such an interesting manuscript. The topic is of importance. I hope you are able to continue investigating gambling among adolescents and young adults and study interventions to address this addiction.

As I read your manuscript, I found grammatical errors related to punctuation, specifically the absence of commas before the expression such as when giving a list of examples. I suggest you delete the expression that said at the beginning of sentences in a professional journal. I found the first paragraph under subheadings 3.2.6 and 3.3 a little confusing. At the end of page 8, line 306, the statement, giving gambling cognitiondid not make sense to me.

I would not consider the manuscript a systematic review of the literature. I would say you conduct a traditional review of the literature. For example, the synthesis of existing research is more qualitative and it is not based on the quality of studies. I did not see a systematic assessment  of the risk of bias individual studies and overall quality of evidence.

Round 2

Reviewer 1 Report

The authors have responded and modified the paper as requested. Congratulations, good work!